# Beyond Carcinoembryonic Antigens: The Role of CA-125 and CA-199 in Predicting Prognosis of Lung Adenocarcinoma

**DOI:** 10.3390/cancers17091517

**Published:** 2025-04-30

**Authors:** Yen-Kun Ko, Yi-Ting Liu, Chia-Yu Tai, Pang-Hung Hsu, Wei-Ke Kuo

**Affiliations:** 1Division of Respiratory Therapy and Chest Medicine, Department of Internal Medicine, Sijhih Cathay General Hospital, New Taipei 221, Taiwan; ykko33@cgh.org.tw; 2Nursing Department, Sijhih Cathay General Hospital, New Taipei 221, Taiwan; d51aa@cgh.org.tw; 3Radiology Department, Taipei Cathay General Hospital, Taipei 106, Taiwan; cgh347547@cgh.org.tw; 4Department of Bioscience and Biotechnology, National Taiwan Ocean University, Keelung 202, Taiwan; phsu@ntou.edu.tw; 5Center of Excellence for the Oceans, National Taiwan Ocean University, Keelung 202, Taiwan

**Keywords:** lung cancer, serum tumor markers, CA125, CA153, CA199, prognosis

## Abstract

Lung adenocarcinoma is a common type of lung cancer in Taiwan, where patients often have a unique genetic profile, including a high rate of epidermal growth factor receptor (EGFR) mutations. While carcinoembryonic antigens (CEAs) are widely used as a tumor marker, the role of other markers, such as CA-125, CA-153, and CA-199, in predicting disease progression is unclear. This study analyzed data from over 1100 patients across multiple hospitals to evaluate these markers. The results showed that CA-125 and CA-199 are better than CEAs at predicting worse outcomes in advanced lung adenocarcinoma, particularly in patients receiving targeted therapy for EGFR mutations. Since these markers are already available in clinical practice, they could help doctors make better treatment decisions. However, further research is needed to confirm these findings and explore how they can be used to improve patient care.

## 1. Introduction

Lung cancer is the leading cause of cancer-related death worldwide [1]. Despite significant efforts and advancements in diagnosis and treatment, overall survival (OS) rates remain poor [2]. In Taiwan, when compared to global trends, lung adenocarcinoma exhibits notable differences, with a notably higher prevalence and unique clinical characteristics. Furthermore, it accounts for 60–70% of all lung cancer cases in Taiwan, significantly exceeding the global average of 40–50% [3]. A significant proportion of these patients are women, even among non-smokers, highlighting a distinct demographic pattern [4]. Additionally, the epidermal growth factor receptor (EGFR) mutation rate in Taiwanese lung adenocarcinoma patients is remarkably high, at 50–60%, compared to the 10–20% observed in Western populations [5]. To enable personalized treatment, improve prognosis accuracy, and guide more effective clinical decision-making, we need biomarkers to predict lung adenocarcinoma outcomes.

Serum tumor markers (STMs) are substances, often proteins, produced by cancer cells or the body in response to cancer. They can be detected in the blood, tissue, or other bodily fluids, and their levels can provide important information about the presence and behavior of malignancies [6]. The most commonly used and most evident STM in lung adenocarcinoma is the carcinoembryonic antigen (CEA) [7]. Elevated CEA levels have been associated with poor prognosis and can be used to track disease progression in patients undergoing therapy [8]. Despite their potential benefits, the predictive accuracy of CEAs can be limited due to biological variability and differences in tumor biology among patients [9].

CA-125, a high-molecular-weight glycoprotein originally associated with ovarian cancer, has been reported to be expressed in some lung adenocarcinomas and may reflect tumor burden or pleural involvement in cancers [10]. CA-153, a mucin-related antigen primarily used in breast cancer, is occasionally elevated in mucin-producing lung tumors [11]. CA-199, derived from gastrointestinal epithelial cells, may be detected in pulmonary tissue and bronchial secretions; an elevation in its levels in lung cancer may reflect tumor progression or metastasis [12]. These STMs are widely used and easily accessible in clinical practice [13]; however, they have not been routinely used in lung cancer management [6]. Most prior investigations have focused on their diagnostic or staging relevance in other cancer types, with few large-scale studies evaluating their predictive significance in lung adenocarcinoma [14]. Comparisons with CEAs are even rarer, and the conclusions of the existing studies are inconsistent [15]. Therefore, we aimed to evaluate their clinical utility in a real-world cohort of patients with lung adenocarcinoma in Taiwan.

We propose that the CA-125, CA-153, and CA-199 STMs may aid in diagnosis and prognosis prediction. This study retrospectively evaluated patients with lung adenocarcinoma treated in our institutions via chart review. We aimed to determine the clinical benefit of STMs that are not commonly used, but are easily accessible in patients with lung adenocarcinoma.

## 2. Materials and Methods

This retrospective multicenter study was conducted at three branches of the Cathay General Hospital network in Taiwan—Taipei, Hsinchu, and Sijhih. We included adult patients (aged 18 years and older) who were diagnosed with pathology-confirmed lung adenocarcinoma between January 2014 and December 2021. The included patients were treated and followed up in our hospitals regarding data for at least one of these four STMs: CEA, CA-125, CA-153 and CA-199. The study protocol received approval from the Institutional Review Board of Cathay General Hospital (IRB No. CGH-P111043) and was carried out in compliance with the ethical principles outlined in the Declaration of Helsinki. Owing to the retrospective nature of the study, the requirement for informed patient consent was waived by the ethics committee.

Data recorded at the point of initial diagnosis were extracted from patient medical charts as follows: age, sex, initial stages, initial treatment type, and STMs (including CEA, CA-125, CA-153, and CA-199). Due to the relatively high levels of STMs and significant interindividual differences, the laboratory parameters were logarithmized.

Data analysis was carried out using SPSS (version 22; SPSS Inc., Chicago, IL, USA). We tested the linear trend in the distribution of tumor marker levels across cancer stages using linear contrast within the general linear model for continuous variables. The resulting *p* for trend values were used to evaluate whether marker levels increased significantly with advancing disease stage. Disease-free survival (DFS) applies to advanced-stage patients and is defined as the time from initial surgery to disease recurrence or death, while progression-free survival (PFS) applies to early-stage cancer after curative treatment, measuring the time from treatment initiation to disease progression or death [16]. Cox proportional hazards regression with backward stepwise selection was used to determine independent predictors of survival. Variables demonstrating statistical significance (*p* < 0.05) in univariate analysis, as well as age and sex, were included in the multivariate model. A complete-case analysis approach was used, in which only patients with complete data for all included covariates were entered into the multivariate model to avoid bias due to missing data [17]. Statistical significance was set at *p* < 0.05, and results were reported as hazard ratios (HRs) with 95% confidence intervals (Cis).

## 3. Results

### 3.1. Demographic Data

The characteristics of the 1133 included patients are summarized in Table 1. The most diagnosed stage among the cohort was stage IV, followed by stage IA. The most administered treatment was surgery, followed by tyrosine kinase inhibitors (TKIs) and platinum-doublet chemotherapy.

### 3.2. Tumor Markers and Clinical Stages

The relationship between different STMs and stages of lung adenocarcinoma are shown in Table 2. After the logarithmic transformation of STMs, it was found that CEA, CA-125, CA-153, and CA-199 levels significantly increased with advancing stages. A similar trend was observed in stage IV lung adenocarcinoma patients treated with EGFR TKIs (Appendix A).

### 3.3. Survival Analysis

The median duration of follow-up for survivors was 1.9 years. Table 3 shows the results of the univariate and multivariate analyses for predictors of DFS in patients with stage I and II adenocarcinoma who underwent surgery of curative intent. Age, sex, and the logarithm of CEA were significant in the univariate analysis and were entered into the multivariate model. In the multivariate analysis, the logarithm of CEA was an independent predictor for DFS (HR = 2.58, 95%CI = 1.80–3.71, *p* < 0.001). Table 4 presents the results of the univariate and multivariate analyses of the predictors for PFS among patients with stage IV adenocarcinoma. The logarithmically transformed values for CEA, CA-125, and CA-199 demonstrated significance in the univariate analysis. These variables, along with age and sex, were subsequently included in the multivariate model. In the multivariate analysis, the logarithm of CA-125 and CA-199 emerged as independent predictors for PFS (HR = 1.17, 95%CI = 1.05–1.30, *p* = 0.004 and HR =1.09, 95%CI = 1.00–1.18, *p* = 0.049, respectively).

Table 5 displays the findings of the univariate and multivariate analyses of predictors for PFS among patients with stage IV adenocarcinoma treated with EGFR TKIs. In the univariate analysis, the logarithmic transformations of CEA and CA-125 demonstrated significance; thus, they were included alongside age and sex in the multivariate model. In the multivariate analysis, the logarithm of CA-125 emerged as an independent predictor for PFS (HR = 1.33, 95%CI = 1.17–1.51, *p* < 0.001).

## 4. Discussion

In summary, the principal findings suggest that firstly, CA-125, CA-153, and CA-199, increased with advancing stages of lung adenocarcinoma patients; secondly, in patients with stage IV adenocarcinoma, higher CA-125 and CA-199 levels may predict poor PFS; and thirdly, higher CA-125 levels were associated with poor PFS in patients with stage IV adenocarcinoma receiving EGFR TKIs. These easily obtainable STMs might thus be helpful in clinical practice, especially for predicting prognosis.

CA-125, an inhibitor of natural killer cells, is commonly used in the treatment of ovarian cancer. In the context of lung cancer, there is substantial evidence that human lung cancer cells can produce CA-125 in vitro [10]. Research by Yuanyuan et al. [18] suggests that ERO1L may influence CA-125 secretion through the IL6 signaling pathway, creating a positive feedback loop that could further drive lung cancer development. Clinically, CA-125 has been reported to predict immunotherapy response in patients with metastatic NSCLC [19], the treatment efficacy of crizotinib and alectinib in patients with ALK-positive NSCLC [20], and the survival of patients with stable disease receiving a combination of immunotherapy, chemotherapy, and anti-angiogenesis therapy in stage IV or recurrent metastatic non-squamous NSCLC [21]. Furthermore, the CA-125 serum level was considered an important indicator of lung cancer with liver metastasis [22]. In our study, the multivariate analysis showed that CA-125, rather than CEA, could predict poor PFS in patients with stage IV adenocarcinoma, as well as those receiving TKIs. Unlike CEA, which showed limited predictive value in this group, CA-125 may capture aspects of tumor biology that are not fully represented by traditional markers.

CA-153, a soluble form of mucin-1 released into the serum, may be associated with mucin-1 expression in nonsquamous cell carcinoma tissue [11]. Our research indicated that CA-153 did not have prognostic predictive values, and these results are consistent with a review article by Trulson et al. [15] reporting that, in most studies, CA-153 failed to predict the outcomes of patients with lung cancer. However, similarly to CEA, CA-125, and CA-199, CA-153 still increased with the lung cancer stage. Chen et al. [14] revealed that CA-153 level correlated with the T stage, N stage, and M stage of lung adenocarcinoma, and the M stage of lung squamous cell carcinoma. Our findings reinforce the interpretation that CA-153 may serve as a staging-associated marker rather than a reliable outcome predictor.

CA-199, primarily used in the diagnosis and management of pancreatic carcinoma, is thought to facilitate the movement of cancer cells from the bloodstream to distant organs [23]. Elevated levels of CA-199 in the blood suggest its production and release by cancer cells. This antigen is expressed in normal epithelial tissues of the pancreas, gallbladder, stomach, bronchus, ovary, and fallopian tubes [24]. Additionally, CA-199 is synthesized and secreted by normal epithelial cells in the central airways and respiratory glands [12]. Our study suggests that CA-199 has a better ability to predict PFS than CEA in patients with stage IV lung adenocarcinoma; however, the opposite was true for early-stage patients, where CA-199 could not predict PFS, while CEA could. These results are generally in line with previous research by Kawai et al. [25], wherein serum CA-199 was associated with survival in advanced lung adenocarcinoma, but not in early-stage cases. In malignant diseases, hypoxia accelerates tumor proliferation and CA-199 production, with cell damage and neovascularization further increasing the release of CA-199 into the bloodstream. This explains why serum CA-199 levels are challenging to use in early-stage lung cancer, as CA-199 release may not be significantly elevated at the initial stages of tumor development [26].

In our study, CEA could better predict the DFS of patients with early-stage adenocarcinoma who underwent surgery of curative intent than other STMs; however, the ability to predict PFS in patients with stage IV lung adenocarcinoma was inferior to that of CA-125 and CA-199. These findings are generally consistent with the meta-analysis by Wang et al. [26], wherein elevated preoperative serum CEA levels were linked to a poor OS, and the review by Trulson et al. [15], where they reported that most studies found no significant evidence supporting the prognostic relevance of CEA in patients with later-stage NSCLC.

This study had several limitations. First, the retrospective nature of the study led to incomplete data collection, including OS. Second, the population was small and highly heterogeneous, comprising patients at different stages and receiving various therapies, which may have introduced significant bias and affected the outcomes. Third, some demographic data were missing, potentially impacting the results, including information on biomarkers, sites of metastasis, PD-L1 status, functional status, and more.

## 5. Conclusions

In conclusion, our study results support the use of seldom-used but easily accessible STMs to evaluate prognosis in patients with lung adenocarcinoma. Higher levels of CA-125 and CA-199 may be more effective than CEA in predicting poor PFS in patients with stage IV adenocarcinoma, with CA-125 demonstrating particular effectiveness in those who received EGFR TKIs. Further large prospective studies are warranted.

## Figures and Tables

**Table 1 cancers-17-01517-t001:** Characteristics of patients.

Clinical Features		Number (%)
Gender		
	Male	446 (39.4)
	Female	687 (60.6)
Age		62.4 ± 12.4
Stage		
	IA	462 (40.8)
	IB	72 (6.4)
	IIA	5 (0.4)
	IIB	11 (1.0)
	IIIA	32 (2.8)
	IIIB	24 (2.1)
	IV	527 (46.5)
Treatment type		
	Tyrosine kinase inhibitor	351 (31.0)
	Tyrosine kinase inhibitor + anti-angiogenesis agent	40 (3.5)
	Platinum-doublet chemotherapy	86 (7.6)
	Single agent chemotherapy	34 (3.0)
	Chemotherapy + anti-angiogenesis	18 (1.6)
	Surgery	473 (41.7)
	Surgery + adjuvant/neoadjuvant chemotherapy	81 (7.1)
	Concurrent chemoradiotherapy	14 (1.2)
	Radiotherapy +/− single agent chemotherapy	3 (0.3)
	Immunotherapy +/− chemotherapy	21 (1.9)
	Immunotherapy + platinum-doublet chemotherapy + anti-angiogenesis agent	9 (0.8)

Notes: Data are presented as frequency (%).

**Table 2 cancers-17-01517-t002:** Tumor markers in patients with adenocarcinoma at different stages.

	Stage I and II (N = 550)	Stage III (N = 56)	Stage IV (N = 527)	*p* for Trend
CEA (log)	0.54 ± 0.73 (N = 506)	2.05 ± 1.73 (N = 55)	3.03 ± 2.22 (N = 519)	<0.001
CA125 (Log)	2.81 ± 0.59 (N = 92)	3.91 ± 1.27 (N = 28)	4.39 ± 1.46 (N = 332)	<0.001
CA153 (Log)	2.28 ± 0.46 (N = 45)	2.35 ± 0.48 (N = 2)	3.44 ± 1.30 (N = 68)	<0.001
CA199 (Log)	2.52 ± 0.81 (N = 142)	2.86 ± 1.23 (N = 32)	3.49 ± 1.97 (N = 401)	<0.001

Notes: Data are presented as mean ± standard deviation. Abbreviations: CEA, carcinoembryonic antigen; CA, cancer antigen.

**Table 3 cancers-17-01517-t003:** Univariate and multivariate analyses of predictors for disease-free survival of patients with stage I and II adenocarcinoma who underwent surgery with curative intent (n = 461).

Variables	Univariate		Multivariate	
	HR (95%CI)	*p*-Value	HR (95%CI)	*p*-Value
Age	1.04 (1.00–1.07)	0.029		
Sex	0.50 (0.25–0.99)	0.046		
CEA (Log)	2.58 (1.80–3.71)	<0.001	2.58 (1.80–3.71)	<0.001
CA125 (Log)	1.50 (0.40–5.78)	0.549		
CA153 (Log)	0.14 (0.00–11.51)	0.377		
CA199 (Log)	1.86 (0.81–4.27)	0.147		

Note: Multivariate analysis was based on 394 patients. Abbreviations: HR, hazard ratio; CI, confidence interval; CEA, carcinoembryonic antigen; CA, cancer antigen.

**Table 4 cancers-17-01517-t004:** Univariate and multivariate analyses of predictors for the progression-free survival of patients with stage IV adenocarcinoma (n = 527).

Variables	Univariate		Multivariate	
	HR (95%CI)	*p*-Value	HR (95%CI)	*p*-Value
Age	1.00 (0.99–1.01)	0.411		
Sex	0.70 (0.56–0.86)	0.001		
CEA	1.10 (1.05–1.16)	<0.001		
CA125 (Log)	1.24 (1.13–1.36)	<0.001	1.17 (1.05–1.30)	0.004
CA153 (Log)	1.05 (0.83–1.32)	0.713		
CA199 (Log)	1.10 (1.03–1.18)	0.005	1.09 (1.00–1.18)	0.049

Note: Multivariate analysis was based on 266 patients. Abbreviations: HR, hazard ratio; CI, confidence interval; CEA, carcinoembryonic antigen; CA, cancer antigen.

**Table 5 cancers-17-01517-t005:** Univariate and multivariate analyses of predictors for the progression-free survival of patients with stage IV adenocarcinoma treated with EGFR tyrosine kinase inhibitor (n = 333).

Variables	Univariate		Multivariate	
	HR (95%CI)	*p*-Value	HR (95%CI)	*p*-Value
Age	1.01 (0.99–1.02)	0.430		
Sex	0.80 (0.60–1.07)	0.134		
CEA (Log)	1.12 (1.06–1.20)	<0.001		
CA125 (Log)	1.32 (1.17–1.51)	<0.001	1.33 (1.17–1.51)	<0.001
CA153 (Log)	1.07 (0.83–1.39)	0.586		
CA199 (Log)	1.11 (0.99–1.24)	0.069		

Note: Multivariate analysis was based on 178 patients. Abbreviations: EGFR, epidermal growth factor receptor; HR, hazard ratio; CI, confidence interval; CEA, carcinoembryonic antigen; CA, cancer antigen.

## Data Availability

The datasets used and analyzed in the current study are available from the corresponding author on reasonable request.

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
