# Peer review of "Beyond Carcinoembryonic Antigens: The Role of CA-125 and CA-199 in Predicting Prognosis of Lung Adenocarcinoma"

_cancers, 2025, doi:10.3390/cancers17091517_

Round 1
Reviewer 1 Report
Comments and Suggestions for Authors
Dear Authors
This manuscript extensively collected data from more than one thousand patients with lung adenocarcinoma, and counted common but not commonly used serum tumor markers in lung adenocarcinoma, including CA-153, CA-125, CA-199, etc. This is a bold attempt, especially the analysis of the correlation between these indicators and the effects of dartboard therapy, as well as the pointing out of the relationship between changes in indicators and patient prognosis, which is of substantial help in the tracking of lung adenocarcinoma treatment and is very impressive. There are a few questions that I would like the authors to supplement and adjust:
- When collecting data on the stages of patients, the focus is on the early stage (stage 1) and the late stage (stage IV), which is very different from the normal distribution we know. Is it related to the national policy promoting early screening (LDCT) so that a large number of early lung cancers can be screened? In the early years, lung adenocarcinoma was often not discovered until the late stage when symptoms appeared, resulting in the phenomenon that there were more cases at the two ends and fewer cases in the middle stages in the data?
- This is a writing suggestion. It is customary that abbreviations of proper nouns will not appear in the abstract. They only need to be marked when they appear for the first time in the text and listed in the abbreviation table for the convenience of readers. If the abbreviation is already commonly used in the field, use the abbreviation directly in the abstract and the main text.
Author Response
Comments 1: When collecting data on the stages of patients, the focus is on the early stage (stage 1) and the late stage (stage IV), which is very different from the normal distribution we know. Is it related to the national policy promoting early screening (LDCT) so that a large number of early lung cancers can be screened? In the early years, lung adenocarcinoma was often not discovered until the late stage when symptoms appeared, resulting in the phenomenon that there were more cases at the two ends and fewer cases in the middle stages in the data?
Response 1:
Thank you for your thoughtful comment regarding the distribution of clinical stages in our cohort. As you noted, there is a notable predominance of stage IA and stage IV cases, with relatively fewer patients in intermediate stages. This pattern appears atypical compared to a standard distribution; however, it aligns with recent epidemiological trends in Taiwan and may be explained by both national and institutional factors.
According to the Taiwan Cancer Registry, the overall stage distribution among lung cancer patients in recent years (2021) shows approximately 5.4 % in stage 0, 29,9 % in stage I, 3.4 % in stage II, 11.0 % in stage III, and 50.2% in stage IV. These highlight that a significant proportion of patients are still diagnosed at advanced stages, particularly stage IV, due to delayed symptom-driven diagnosis.
In contrast, our study population shows a notably higher proportion of stage I patients (40.8%), which may be attributed to two main reasons:
- Impact of early detection programs, such as the nationwide promotion of low-dose computed tomography (LDCT) screening, especially in high-risk groups, which has increased the detection of asymptomatic early-stage cancers.
- Institutional referral pattern: Our hospital is widely recognized for its thoracic surgery expertise, which attracts a substantial number of early-stage lung cancer patients from outside hospitals who seek surgical intervention. This referral bias likely contributes to the elevated proportion of stage I cases in our dataset.
Comments 2: This is a writing suggestion. It is customary that abbreviations of proper nouns will not appear in the abstract. They only need to be marked when they appear for the first time in the text and listed in the abbreviation table for the convenience of readers. If the abbreviation is already commonly used in the field, use the abbreviation directly in the abstract and the main text.
Response 2:
Thank you very much for your helpful writing suggestion regarding the use of abbreviations in the abstract. We appreciate the reminder about the convention that abbreviations of proper nouns are typically defined only upon their first appearance in the main text, and not necessarily within the abstract.
In our revision, we have carefully reviewed the use of abbreviations such as EGFR, STM, CEA, CA-125, CA-153, CA-199, DFS, and PFS in the abstract. Since EGFR and CEA are widely recognized abbreviations in the field of oncology, we have retained their usage in the abstract without redefinition. However, for less commonly used abbreviations (e.g., DFS, PFS, STM, TKI), we have followed the conventional format and either spelled them out or adjusted their usage accordingly.
We made changes in
- Page 1, line: 25 : epidermal growth factor receptor (EGFR) mutations à EGFR mutations
- Page 1, line 25: carcinoembryonic antigen (CEA) à CEA
- Page 1, line 34.35: TKIs à tyrosine kinase inhibitors (TKIs)
Reviewer 2 Report
Comments and Suggestions for Authors
The manuscript entitled “Beyond CEA: Role of CA-125 and CA-199 in Predicting Prognosis of Lung Adenocarcinoma” presents the results of a multicenter, retrospective review study including data from 1133 patients treated for lung adenocarcinoma. In this study, the Authors compared the prognostic value of serum tumor markers: carcinoembryonic antigen (CEA) which is used mainly in lung adenocarcinoma (to track disease progression in patients undergoing therapy) in comparison to CA-125, CA-153, and CA-199 that are widely used in the diagnosis and monitoring of ovarian, breast cancer, and pancreatic cancers.
The study is well-designed, and the methods and results are described in detail. The information in tables and captions is adequate.
Major comments
The introduction is brief, describing the use of CEA in lung cancer more extensively, while the rest of the STMs analyzed are described together in one sentence. The authors should devote at least one small paragraph to each of the analyzed serum proteins in the introduction to indicate to the reader why they chose these STMs for their study. Please also include information on the studies analyzing the CA-125, CA-153, and CA-199 in lung cancer, as it currently appears that there were no significant prior investigations in lung cancer.
In the Materials and Methods section, please provide information about the size of the group in which more biomarkers were analyzed together (CA-125 /CA-153/CA- 199). Were there patients in whom all 4 biomarkers were analyzed? For this group, were the results consistent with the results for the entire analyzed cohort?
Did you also have access to the TNM staging for patient clinic-pathology reports? Do the CA-125 /CA-153/CA-199 serum levels correlate with node involvement or cancer metastasis?
Please provide ROC curve analyses to assess the sensitivity and specificity of the studied STMs in estimating their diagnostic value for PFS and DFS.
In Table 2, please provide information on the statistical test used to compare those 3 groups. You could add one column (or a table in the supplement) with data for stage IV treated with an EGFR inhibitor.
The discussion is well written, but in the reviewers' opinion, the authors ' own results should be discussed more broadly. (e.g., lines 164-166; 168-170).
In line 163, there is a mention of the “IMpower150” study – please provide more information regarding this study, or omit the clinical trial name, as you do not refer to clinical studies in other paragraphs.
Author Response
Comments 1 : The introduction is brief, describing the use of CEA in lung cancer more extensively, while the rest of the STMs analyzed are described together in one sentence. The authors should devote at least one small paragraph to each of the analyzed serum proteins in the introduction to indicate to the reader why they chose these STMs for their study. Please also include information on the studies analyzing the CA-125, CA-153, and CA-199 in lung cancer, as it currently appears that there were no significant prior investigations in lung cancer.
Response 1 :
Thank you for your valuable comment. To provide a more comprehensive rationale for our selection of serum tumor markers (STMs), we briefly introduce each marker and review its relevance in lung cancer research. We have re-wrote the paragraph in the section of introduction as follows (page 2, third paragraph, line 64-76):
CA-125, a high-molecular-weight glycoprotein originally associated with ovarian cancer, has been reported to be expressed in some lung adenocarcinomas and may reflect tumor burden or pleural involvementcancers[10]. CA-153, a mucin-related antigen primarily used in breast cancer, is occasionally elevated in mucin-producing lung tumors[11]. CA-199, derived from gastrointestinal epithelial cells, may be detected in pulmonary tissue and bronchial secretions; its elevation in lung cancer may reflect tumor progression or metastasis[12]. These STMs are widely used and easily accessible in clinical practice [13], however, they have not been routinely used in lung cancer management[6]. Most prior investigations have focused on their diagnostic or staging relevance in other cancer types, with few large-scale studies evaluating their predictive significance in lung adenocarcinoma[14]. Comparisons with CEA are even rarer, and the conclusions of the existing studies are inconsistent[15]. Therefore, we aimed to evaluate their clinical utility in a real-world cohort of patients with lung adenocarcinoma in Taiwan.
And the corresponding references was as follows: (Page 7, line 260-261; page 7, line 267-274; page 8, line 275-278)
- Homma, S.; Satoh, H.; Kagohashi, K.; Fujiwara, M.; Kamma, H.; Sekizawa, K. Production of CA125 by human lung cancer cell lines. Clinical and Experimental Medicine 2004, 4, 139-141.
- Situ, D.; Wang, J.; Ma, Y.; Zhu, Z.; Hu, Y.; Long, H.; Rong, T. Expression and prognostic relevance of MUC1 in stage IB non-small cell lung cancer. Medical oncology 2011, 28, 596-604.
- Matsuoka, Y.; Endo, K.; Kawamura, Y.; Yoshida, T.; Saga, T.; Watanabe, Y.; Koizumi, M.; Nakashima, T.; Konishi, J.; Yamaguchi, N. Normal bronchial mucus contains high levels of cancer‐associated antigens, CA125, CA19‐9, and carcinoembryonic antigen. Cancer 1990, 65, 506-510.
- Duffy, M. Clinical uses of tumor markers: a critical review. Critical reviews in clinical laboratory sciences 2001, 38, 225-262.
- Duffy, M.J. Role of tumor markers in patients with solid cancers: a critical review. European journal of internal medicine 2007, 18, 175-184.
- Chen, Z.-q.; Huang, L.-s.; Zhu, B. Assessment of seven clinical tumor markers in diagnosis of non‐small‐cell lung cancer. Disease Markers 2018, 2018, 9845123.
- Trulson, I.; Holdenrieder, S. Prognostic value of blood-based protein biomarkers in non-small cell lung cancer: A critical review and 2008–2022 update. Tumor Biology 2023, 1-51.
Comments 2: In the Materials and Methods section, please provide information about the size of the group in which more biomarkers were analyzed together (CA-125 /CA-153/CA- 199). Were there patients in whom all 4 biomarkers were analyzed? For this group, were the results consistent with the results for the entire analyzed cohort?
Response 2 :
Thank you for your helpful comment. As described in the Materials and Methods section, we applied a standard model selection approach, including only variables with a p-value < 0.05 in the univariate analysis, along with age and sex, in the multivariate Cox regression.
To minimize bias related to missing data, we conducted a complete-case analysis, meaning only patients with complete data across all selected covariates—including the tumor markers—were entered into the multivariate model.
We have clarified the cohort size used in the multivariate analysis in both the Materials and Methods section and the corresponding Tables 3, 4, and 5.
- we added the following statement to the Methods section: (page 3, line 108-110)
A complete-case analysis approach was used, in which only patients with complete data for all included covariates were entered into the multivariate model to avoid bias from missing data[17]
And the corresponding reference:(page 8 , line 281-282)
- Lin, D.; Ying, Z. Cox regression with incomplete covariate measurements. Journal of the American Statistical Association 1993, 88, 1341-1349.
- In addition, we indicated the number of patients included in the multivariate analyses directly in Tables 3, 4, and 5, with the following notation: “Note: multivariate analysis was based on XXX patients.” (page 4, line 146; page 5, line 151; page 5, line 162)
We greatly appreciate your suggestion, which helped improve the clarity and transparency of our methodology.
Comments 3: Did you also have access to the TNM staging for patient clinic-pathology reports? Do the CA-125 /CA-153/CA-199 serum levels correlate with node involvement or cancer metastasis?
Response 3:
Thank you for this valuable question. Unfortunately, detailed TNM staging data such as lymph node involvement (N stage) or distant metastasis (M stage) were not available in our dataset and were therefore not included in our analysis. Our study focused primarily on evaluating the prognostic value of serum tumor markers (CEA, CA-125, CA-153, and CA-199) based on overall stage classification and survival outcomes.
However, previous studies have indicated potential correlations between elevated serum levels of these markers and disease burden. For example:
- CA-125 has been associated with pleural involvement and distant metastasis in advanced lung adenocarcinoma
Kimura, Y., et al. "Serum CA125 level is a good prognostic indicator in lung cancer." British journal of cancer 62.4 (1990): 676-678.
- CA-199 has also been linked to advanced tumor stage and metastasis in gastrointestinal and, in some reports, lung cancer
Mumbarkar, P.; Raste, A.; Ghadge, M. Significance of tumor markers in lung cancer. Indian Journal of Clinical Biochemistry 2006, 21, 173-176.
These findings are consistent with our observations that CA-125 and CA-199 are more predictive of progression-free survival in stage IV patients.
We acknowledge that further integration of TNM-specific data may provide additional insights and could be addressed in future studies.
Comments 4: Please provide ROC curve analyses to assess the sensitivity and specificity of the studied STMs in estimating their diagnostic value for PFS and DFS.
Response 4:
Thank you for your constructive suggestion. In response, we conducted additional ROC curve analyses to evaluate the discriminatory performance of the studied serum tumor markers.
Specifically, we assessed 5-year disease-free survival (DFS) in early-stage patients and 1-year progression-free survival (PFS) in stage IV patients, calculating the corresponding area under the ROC curve (AUC) as well as sensitivity and specificity at various cutoff values. However, the results of these analyses were suboptimal, with AUC values failing to demonstrate strong discriminative power in either group.
These findings may be due to the time-dependent and censored nature of survival outcomes, which traditional ROC curves do not fully account for. As such, we believe that time-to-event analysis using Cox proportional hazards models remains a more appropriate and informative approach for evaluating prognostic biomarkers in this context.
Nonetheless, we greatly appreciate your suggestion and remain open to including time-dependent ROC curves as supplementary material if deemed helpful.
5 year disease-free survival of patients with stage I and II adenocarcinoma who underwent surgery with curative intent
1-year progression-free survival of patients with stage IV adenocarcinoma
1-year progression-free survival of patients with stage IV adenocarcinoma treated with EGFR tyrosine kinase inhibitor
Comments 5: In Table 2, please provide information on the statistical test used to compare those 3 groups. You could add one column (or a table in the supplement) with data for stage IV treated with an EGFR inhibitor.
Response 5:
Thank you for your valuable suggestions.
We have revised the third paragraph of the Materials and Methods section to clarify the statistical approach used in Table 2. The sentence:
“We tested the linear trend of the distribution of the levels of tumor markers across different cancer stages using linear contrast in the general linear model for continuous variables.”
has been revised to: (page 3, line 98-101)
“We tested the linear trend in the distribution of tumor marker levels across cancer stages using linear contrast within the general linear model for continuous variables. The resulting p for trend values were used to evaluate whether marker levels increased significantly with advancing disease stage.”
In addition, in response to your recommendation, we conducted a subgroup analysis among stage IV lung adenocarcinoma patients treated with EGFR TKIs. The results, presented in Supplementary Table 1, demonstrated a similar trend in STM levels as seen in the full cohort, supporting the robustness of our findings.
Supplement Table 1. Tumor markers in patients with adenocarcinoma at different stages
|
Stage I and II (N = 550) |
Stage III (N = 56) |
Stage IV treated with EGFR TKI (N = 527) |
P for trend |
CEA (log) |
0.54 ± 0.73 (N = 506) |
2.05 ± 1.73 (N = 55) |
3.03 ± 2.18 (N = 319) |
<0.001 |
CA125 (Log) |
2.81 ± 0.59 (N = 92) |
3.91 ± 1.27 (N = 28) |
4.35 ± 1.47 (N = 213) |
<0.001 |
CA153 (Log) |
2.28 ± 0.46 (N = 45) |
2.35 ± 0.48 (N = 2) |
3.38 ± 1.39 (N = 51) |
<0.001 |
CA199 (Log) |
2.52 ± 0.81 (N = 142) |
2.86 ± 1.23 (N = 32) |
3.35 ± 1.72 (N = 252) |
<0.001 |
Notes: Data are presented as mean ± standard deviation.
Abbreviations: EGFR, epidermal growth factor receptor; TKI, tyrosine kinase inhibitor; CEA, Carcinoembryonic antigen; SCC, Squamous cell carcinoma antigen; CA, Cancer antigen
We also added the following sentence in the Results section (page 4, line 125-127) to reflect this addition:
“A similar trend was observed in stage IV lung adenocarcinoma patients treated with EGFR TKIs (Supplementary Table 1).”
Comments 6: The discussion is well written, but in the reviewers' opinion, the authors ' own results should be discussed more broadly. (e.g., lines 164-166; 168-170).
Response 6:
Thank you for your insightful suggestion. In response, we have expanded the discussion of our own findings regarding CA-125 and CA-153 to better contextualize them in relation to existing literature. Specifically, we added one interpretative sentence to each paragraph to highlight the clinical implications of our results and to distinguish their prognostic versus staging relevance. These additions help clarify how our data contribute to the current understanding of STM utility in lung adenocarcinoma.
CA-125: (page 6, line 184-186)
Unlike CEA, which showed limited predictive value in this group, CA-125 may capture aspects of tumor biology not fully represented by traditional markers.
CA-153: (page 6, line 194-195)
Our findings reinforce the interpretation that CA-153 may serve as a staging-associated marker rather than a reliable outcome predictor.
We hope these enhancements address the reviewer’s comment and improve the clarity and depth of our discussion.
Comments 7: In line 163, there is a mention of the “IMpower150” study – please provide more information regarding this study, or omit the clinical trial name, as you do not refer to clinical studies in other paragraphs.
Response 7:
Thank you for your helpful comment. In response, we removed the mention of the clinical trial name “IMpower150” to maintain consistency with the rest of the manuscript, as other clinical trials were not specifically named. To clarify the clinical context, we revised the sentence to: (page 6, line 179-181)
, and the survival of patients with stable disease receiving a combination of immunotherapy, chemotherapy, and anti-angiogenesis therapy in stage IV or recurrent metastatic non-squamous NSCLC.
We believe this phrasing retains the key clinical implication while improving clarity and consistency.
Round 2
Reviewer 2 Report
Comments and Suggestions for Authors
The updated version of the manuscript's introduction, as presented, provides a clearer explanation for the selection of CA-125, CA-153, and CA-199. The Materials and Methods section is also more detailed, particularly in its broader description of the statistical analysis.
The additional statistical analysis performed and presented in the response to the review may enhance the quality and content of this manuscript. It is positive that the subgroup analysis of EGFR TKIs-treated AC patients (with stage IV) demonstrated a similar trend in STM levels as seen in the entire cohort, which lends more credibility to the results.
In the Reviewer's opinion, including the ROC curves in the final version of the supplementary methods would be beneficial, even if those results are suboptimal, given that the AUC values fail to demonstrate strong discriminative power in either group.
I agree that analysis using Cox proportional hazards models reveals better prognostic biomarker properties for the selected CA. Nevertheless, including the ROC/AUC analysis, which is frequently performed in this type of study, broadens the analysis and underscores the importance of this approach.